



# Impact of the ice thickness distribution discretization on the sea ice concentration variability in the NEMO3.6-LIM3 global ocean–sea ice model

Eduardo Moreno-Chamarro[1*], Pablo Ortega[1], and François Massonnet[2]

[1]Barcelona Supercomputing Center (BSC), Barcelona, Spain.
[2]Georges Lemître Centre for Earth and Climate Research, Earth and Life Institute, Université catholique de Louvain, Louvain-la-Neuve, Belgium

*Correspondence to*: Eduardo Moreno-Chamarro (eduardo.moreno@bsc.es)

**Abstract.** This study assesses the impact of different sea ice thickness distribution (ITD) configurations on the sea ice
concentration (SIC) variability in ocean-standalone NEMO3.6-LIM3 simulations. Three ITD configurations with different
numbers of sea ice thickness categories and boundaries are evaluated against three different satellite products (hereafter
referred to as "data"). Typical model and data interannual SIC variability is characterized by k-means clustering both in the
Arctic and Antarctica between 1979 and 2014 in two seasons, January–March and August–October, when coherence across
clusters in individual months is largest. Analysis in the Arctic is done before and after detrending the series with a 2nd degree
polynomial to separate interannual from longer-term variability.

Before detrending, winter clusters capture SIC response to atmospheric variability at both poles and summer cluster a
positive and negative trend in the Arctic and Antarctic SIC respectively. After detrending, Arctic clusters reflect SIC response
to interannual atmospheric variability predominantly. Model–data cluster comparison suggests that no specific ITD
configuration or category number increases realism of the simulated Arctic and Antarctic SIC variability in winter. In the
Arctic summer, more thin-ice categories decrease model–data agreement without detrending but increase agreement after
detrending. Overall, a single-category configuration agrees the worst with data.

Direct model–data comparison of SIC anomaly fields shows that more thick-ice categories improve winter SIC variability
realism in Central Arctic regions with very thick ice. By contrast, more thin-ice categories reduce model–data agreement in
the Central Arctic in summer, due to an overly large simulated sea ice volume.

In summary, whereas better resolving thin ice in NEMO3.6-LIM3 can hamper model realism in the Arctic but improve it
in Antarctica, more thick-ice categories increase realism in the Arctic winter. And although the single-category configuration
performs the worst overall, no optimal configuration is identified. Our results suggest that no clear benefit is obtained from
increasing the number of sea ice thickness categories beyond the current usual standard of 5 categories in NEMO3.6-LIM3.

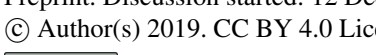



## 1 Introduction

Analyses of recent observations have allowed identifying different drivers of sea ice variability. For example, interannual sea ice variability has primarily been related to changes in atmospheric and oceanic circulation: atmospheric variability, which can directly be related to large-scale atmospheric modes such as the North Atlantic Oscillation (NAO) or Siberian High in the Northern Hemisphere, and the Southern Annular Mode over Antarctica, can drive changes in the sea ice both dynamically and thermodynamically [e.g., Rigor et al., 2002; Rigor and Wallace, 2004; Ogi et al., 2007; Yuan and Li, 2008;

Wang et al., 2009; Hobbs and Raphael, 2010; Holland and Kwok, 2011; Renwick et al., 2012; Kohyama and Hartmann, 2016; Lynch et al., 2016; Close et al., 2017; Blackport et al., 2019; Olonscheck et al., 2019]. Similarly, interannual changes in ocean heat transport to high latitude can contribute to anomalous episodes of Arctic sea ice melting in both the Atlantic and Pacific sectors [e.g., Hibler, 1986; Venegas and Mysak, 2000; Ingvaldsen et al., 2004a; 2004b; Woodgate et al., 2010; Schlichtholz, 2011]. On longer time scales, the accelerating thinning in Arctic sea ice [Comiso et al., 2008; Serreze and

Stroeve, 2015] might be modulated by lower-frequency variability in modes like the NAO [e.g., Delworth et al., 2016] or Atlantic Multidecadal Variability [e.g., Day et al., 2012; Drinkwater et al., 2014; Miles et al., 2014]. Accurately capturing this complex range of variability in sea ice, together with its potential impacts on lower latitude climate [e.g., Screen, 2013], demands for a realistic representation of the sea ice in climate models.

One among the many crucial features of sea ice to ensure its realistic representation is its thickness complexity, which

determines other important physical properties, such as ice's salt and heat content, resistance to deformation and fracture, and melting and growth rates. State-of-the-art sea ice models typically use an ice thickness distribution (ITD) [Thorndike et al., 1975] to account for subgrid-scale variability of ice properties. In most cases, through an ITD the different ice thicknesses are sorted into a fixed number of categories in a configuration which usually presents the finest resolution in the thinnest ice. Several studies have explored the advantages of including an ITD to simulate the mean state and seasonality in

sea ice accurately, as well as the number of categories that would render the most realistic ice representation [e.g., Bitz et al., 2001; Holland et al., 2006; Massonnet et al., 2011; Uotila et al., 2017; Ungermann et al., 2017; Massonnet et al., 2019]. These studies, however, have partly overlooked the impact of the ITD on the simulated sea ice variability. To our knowledge, only Massonnet et al. [2011] reported a more realistic interannual variability in the Arctic sea ice extent (SIE) in the LIM3 sea-ice model than in its previous model version, LIM2 (although this improvement cannot exclusively be attributed to the

addition of an explicit 5-category ITD in LIM3 but to all the refinements in sea ice parametrizations absent in LIM2). Thus the question of whether a particular ITD configuration or number of categories ensures a more realistic sea ice variability and long-term trend remains unanswered.

Sea ice concentration (SIC) and thickness are the main quantities used to characterize its variability. Most of the previous studies have focused on the impact of an ITD on the sea ice thickness, especially in the Arctic [e.g., Holland et al., 2006;

Hunke, 2014; Ungermann et al., 2017]. By contrast, SIC has received less attention, perhaps motivated by the relatively minor or only indirect effect that the ITD appears to have on the representation of its mean state [e.g., Massonnet et al., 2011; Uotila et al., 2017; Massonnet et al., 2019]. However, while SIC has continuously been measured by satellites since 1978





[Cavalieri et al., 1996; EUMETSAT, 2015], equivalent measurements of thickness have only become available in the past decade [e.g., Laxon et al., 2013]. Literature exploring the observed SIC variability is therefore much richer than that on sea

ice thickness and offers a more exhaustive account of its key features and drivers (see most of the references above). This study therefore represents a step forward with respect to previous ones, as it presents the, to our knowledge, the first detailed assessment of the impact of the ITD discretization on the SIC variability at both poles since 1978, using the state-of-the-art coupled ocean–sea ice model NEMO3.6-LIM3. This study is a companion paper to Massonnet et al. [2019], in which the response of the modelled sea ice climatology to an ITD discretization is investigated.

The paper is structured as follows: Section 2 describes the model and experimental design, Section 3 follows with the main results of the model–data comparison, and Sections 4 finishes with the discussion of the results and main conclusions.

## 2 Model and experimental setup

### 2.1 Model description

We use the dynamic-thermodynamic sea ice model LIM3.6 (Louvain-la-Neuve sea Ice Model) [Rousset et al., 2015]

coupled to a finite-difference, hydrostatic, free-surface-primitive-equation ocean model within the version 3.6 of the NEMO framework (Nucleus for European Modelling of the Ocean) [Madec, 2008]. Only a short description of the model is provided in the following; for more details we refer to Barthélemy et al. [2018] and Massonnet et al. [2019]. Both the ocean and sea ice models are run on the global eORCA1 grid with a 1° nominal zonal resolution. The ocean has 75 vertical levels which increase non-uniformly from 1 m at the surface to 10 m at 100 m depth and 200 m at the bottom. To avoid spurious model

drift, a weak restoring toward the World Ocean Atlas 2013 surface salinity climatology [Zweng et al., 2013] is applied with a strength of 167 mm/day. The restoring is damped under the sea ice (multiplied by one minus its concentratio), where observations are less reliable, to avoid altering ocean–ice interactions.

### 2.2 Experimental setup: atmospheric forcing

The model is run over the period 1979–2014. The atmospheric forcing is provided by the DRAKKAR Forcing Set

version 5.2 (DFS5.2) [Brodeau et al., 2010; Dussin et al., 2016]. This global forcing set is derived from the ERA-Interim reanalysis over the period 1979–2015. It has a spatial resolution close to 0.7°, or 80 km, and it is used within the CORE forcing formulation of NEMO, which uses bulk formulas developed by Large and Yeager [2004]. Continental freshwater inputs include river runoff rates from the climatological dataset of Dai and Trenberth [2002] north of 60°S, prescribed meltwater fluxes from ice shelves along the coastline of Antarctica [Depoorter et al., 2013], and climatological freshwater

fluxes from iceberg melting at the surface of the Southern Ocean [Merino et al., 2016]. Forcing the NEMO3.6-LIM3 model with observation-based atmospheric variability ensures that simulated SIC variability follows observations to a large extent, in particular the atmospheric-driven changes; this allows us to compare model and observations (hereafter also referred as to data) and evaluate the impact of the different ITD configurations.

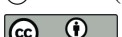



## 2.3 Experimental setup: ITD configurations

LIM3.6 employs an ITD to represent the subgrid-scale distribution of the sea ice thickness, enthalpy, and salinity [Thorndike et al., 1975], discretized into a fixed number of categories. An ITD is characterized by both the number of categories and the position of their boundaries. We run three different sets of sensitivity experiments to evaluate the impact of the ITD on the SIC variability (Fig. 1). In the first set (hereafter, S1), the categories are set by the default ITD discretization of LIM, which varies both the position and the resolution of the thickness categories according to the number

of categories, setting the finest resolution to the thinnest ice. In the second set (S2), new thickness categories are successively appended without changing the existing category boundaries, which allows assessing the impact of the thick ice categories. In the third set of experiments, the lower boundary of the thickest category is set as 4 m depth and the ITD resolution is increased or reduced by merging or splitting the existing categories. The upper limit of 4 m corresponds to the maximum thickness that thermodynamic ice growth can sustain in the Arctic [Maykut and Untersteiner, 1971] and therefore allows the

thickest category to host the deformed ice produced in the model. For more details of the ITD and these experiments we refer to Massonnet et al. [2019].

## 2.4 Reference observations

Arctic and Antarctic SIC variability in the model simulations is compared with that from three satellite observational products for the period 1979–2014: OSI SAF (OSI-409/OSI-409-a) [EUMETSAT, 2015], NSIDC-0051 [Cavalieri et al.,

1996], and HadISST v2.2 [Titchner and Rayner, 2014]. Both OSI SAF and NSIDC provide monthly mean SIC since October, 1978; NSIDC, however, lacks a circular sector centered over the North Pole ("pole hole"), where SIC is set as 1. HadISST blends historical sources, such as sea ice charts, with OSI SAF passive microwave data to provide monthly SIC since January 1850, with concentration values between 0 and 0.15 reset as 0 (open water).

## 2.5 K-means clustering

K-means clustering as included in the s2dverification R package [Manubens et al., 2018] is used to characterize interannual SIC variability in model simulations and observations. K-means clustering aims at simultaneously minimizing the Euclidean distance between members of a given cluster and maximizing the distance between centroids of the different clusters [Wilks, 2011]. It is an alternative method of dimension reduction to other, more commonly used, such as principal component analysis. With respect to those, K-means clustering is more robust in a physical sense, it can account for potential

nonlinearities in a climate field [Andeberg, 2014; Hastie et al., 2009], and it does not assume orthogonality or linearity between dominant modes. K-means clustering has successfully been employed to extract atmospheric weather regimes over the North Pacific and North Atlantic [e.g., Michelangeli et al., 1995, Rossow et al., 2005, Coggins et al., 2014], dynamically similar regions of the global ocean circulation [Sonnewald et al., 2019], or variability clusters from the pan-Arctic sea ice thickness [Fučkar et al., 2015, 2018]. In our case, each cluster is characterized by a pattern of SIC anomalies (cluster





centroids) and a discrete time series of occurrence. Both the spatial features of the patterns and their occurrence in time vary

with the computed total number of clusters, K. Cluster validity, characterised by the most robust choice of K, is determined

using 10 indices (namely, Duda–Hart, Ratkowsky–Lance, Ball–Hall, SD, cubic clustering criterion, traceCovW, Rubin,

Beale, Scott, and Marriot), which forms a selection of the 10 computationally fastest ones out of the 30 included in the

NbClust R package [Charrad et al. 2014]; these indices assess both intra-cluster similarity and inter-cluster dissimilarity. We

test K values between 2 and 5 and evaluate the results of K-means clustering with those validity indices. Since this is a very

computationally demanding analysis, we previously reduce the number of degrees of freedom by interpolating the SIC field

from satellite observations onto a 3° horizontal regular grid. For all seasons and observational datasets the optimal K (i.e.,

the most frequent value for the 10 validity indices) thus evaluated is 3. Therefore we hereafter apply K-means clustering with

K set as 3 to the SIC fields on a 1° horizontal regular grid from both model and observational data. All the calculations are

done over the period 1979–2014. Our results are insensitive to the initial seed used to calculate clusters (not shown).

## 3 Results

### 3.1 Defining the winter and summer seasons

We intend to focus the comparison between simulated and observed SIC variability in two seasons centered around

winter and summer, when maximum and minimum sea ice areas occur respectively. To avoid any *a priori* assumption about

which months define these seasons, we first assess agreement across monthly clusters and aggregate months with similar

variability. Following the steps described in Section 2.3 for each observational product separately, we first calculate 3 (as

optimal number) clusters in each individual month in the Arctic and Antarctica. At each pole, we then compute the spatial

correlation coefficients between all the clusters in any two months. We retain the maximum positive value from the resultant

distribution, which sets the uppermost-limit of cluster agreement between those two months. Results in OSI SAF are shown

in Fig. 2 (results of NSIDC and HadISST are very similar and therefore not shown). Two periods stand out at both poles,

when monthly cluster agreement is largest, January through March (JFM) and August through October (ASO). All the

subsequent analyses focus on these two seasons, which we refer to as winter and summer (even though they include

climatological spring and fall months).

### 3.2 Sea ice extent

Before comparing SIC clusters, we explore the impact of the ITD configuration on the temporal evolution of the Arctic

and Antarctic sea ice extent (SIE) over the period 1979–2014 (Fig. 3). This analysis will help interpret results from the

clusters below. Note that impacts on the simulated climatological mean state and seasonal cycle over this period have

previously been described by Massonnet et al. [2019]. In the model, seasonal SIE is calculated from monthly SIC on the

original model grid; in observations, seasonal SIE is calculated from the monthly SIE directly provided by the different

products. The impact of different ITD configurations on the Arctic SIE in both seasons and Antarctic SIE in winter is





marginal, and all the simulations show values that are within observational uncertainty (which we assume to be defined by the envelope of the different observational products; Fig. 3). The largest differences across simulations are for the summer Antarctic SIE. Increasing the number of categories from 1 to 50 in the S1 configurations reduces the Antarctic SIE by about $4 \cdot 10^6$ km$^2$, although the largest decrease of about $2 \cdot 10^6$ km$^2$ is from S1.01 to S1.03. This renders the simulated SIE values in

S1 runs closer to those in OSI SAF and NSIDC but more different to those in HadISST. HadISST SIE values are consistently above those in OSI SAF and NSIDC in the Arctic and Antarctica in both seasons, as also noted by Titchner and Rayner [2014]. Increasing the number of categories in the S2 and S3 configurations has a comparatively smaller impact, reducing and increasing the summer Antarctic SIE by about $1 \cdot 10^6$ km$^2$ respectively; these results are still within observational uncertainty. The simulated SIE trend is slightly underestimated in the winter Arctic, although it is well captured in summer as

well as in Antarctica in both seasons. In terms of interannual variability, the simulations disagree the most with the observations in Antarctica especially in summer, when simulations show large interannual variations that are not found in observations (for example, around 2000). By contrast, the simulated Arctic SIE variability for all ITD configurations is very close to observations in both seasons.

    To characterize differences between simulated and observed SIC, we calculate the integrated ice edge error (IIEE) as the

total area where model and observations disagree on SIC values above 15% [Goessling et al., 2016]. In general terms, the largest IIEE is in the Arctic and Antarctica in JFM, with the smallest values emerging for the comparison with NSIDC (Supp. Fig. 1). For all the simulations, the IIEE remains relatively constant over the period 1979 –2014 at both poles and seasons, and the impact of a different ITDs on the IIEE is marginal in the Arctic in both seasons and in Antarctica in winter. The situation is different in the Antarctic summer (JFM), when differences in IIEE due to the ITD are the largest (Fig. 4). IIEE

between the simulations and observations is overall larger than across observations for all the ITD configurations. The single-category ITD configuration exhibits the largest IIEE with respect to all the observations. Increasing the number of categories in the S1 and S3 configurations tends to reduce the IIEE by about $1 \cdot 10^6$ km$^2$ between the coarsest and finest resolution. Changes in categories in the S2 configuration have a smaller impact on the IIEE, with no clear improvement or worsening for a finer or coarser ITD. These results suggest that a finer resolution of the thinner ice and not of the thicker ice

to some degree improves the representation of the simulated Antarctic SIC in ASO in our model with respect to observations. This might be related to an improved response of the thin ice (the easiest to melt, grow, and advect) to the atmospheric forcing.

### 3.3 SIC cluster analysis

    In the following, we describe the three clusters of SIC variability in the observations. Clusters in OSI SAF are shown in

Figs. 5 and 6 in the Arctic and Antarctica respectively (since clusters in NSIDC and HadISST are very similar, they are shown in Supp. Figs. 2 and 3 respectively). In the Arctic winter, the first cluster shows four poles of dominant variability, with more ice in the Barents, Greenland, and Okhotsk seas and less ice in the Labrador and Bering seas (Fig. 5); this pattern agrees with the quadrupole mode described by previous literature associated with variations in the strength of the Siberian



High [e.g., Ukita et al., 2007; Close et al., 2017]. The second cluster presents similar centers of action to the first one, but

SIC anomalies are negative in the Labrador, Barents and Okhotsk seas and positive in the Bering Sea. The third cluster shows strong anomalies of opposite sign in the Labrador (strongly positive) and Nordic seas (negative) which resembles the typical fingerprint of a positive NAO phase on the SIC [Bader et al., 2011]. In fact, this cluster dominates between 1990 and 1996, when the winter NAO was persistently positive [Hurrell and Deser, 2010]. Overall, the first and third clusters alternate until 2004 approximately, after which the second cluster dominates. In the last decade, the root mean square distance

between the clusters and the anomaly fields (indicated by the symbol size in Fig. 5) increases to its largest values over the whole period in OSI SAF, but not in NSIDC and HadISST. These results suggest that the winter SIC variability might fundamentally have changed after 2004, in agreement with the observed acceleration in the SIC melting trend [e.g., Comiso et al., 2008; Serreze and Stroeve, 2015].

In the Arctic summer, both the cluster patterns and relative occurrences reflect a long-term melting trend (Fig. 5). The

first and third clusters are very similar, which respectively exhibit widespread positive and negative anomalies in the central Arctic and dominate over the initial period (ca. 1979–1988) and last one (ca. 2005–2014). The second cluster, by contrast, dominates in the middle decades (ca. 1989–2005) and presents a dipole of positive and negative anomalies between the central Arctic and the surroundings. Such partitioning in decades of alternating dominance suggests that the long-term melting trend in sea ice (as seen in the SIE; Fig. 3b) controls the clustering; previously detrending the data might therefore be

necessary for a more robust characterization of the interannual variability (see below).

In Antarctica summer (JFM), the three clusters exhibit poles of dominant variability close to the continental coast, especially in the Weddell and Ross seas (Fig. 6). The first and second clusters show similar patterns but of opposite sign, with an overall decrease or increase respectively but in the Amundsen and Bellinghausen seas. The third cluster shows a dipole of anomalies in the Weddell Sea and positive ones in the Amundsen Sea. Summer SIC variability is dominated by the

first cluster (58%), especially during the first decades. Although the second and especially the third clusters are much less frequent (31% and 11% respectively), the second one tends to dominate in the last decade (ca. 2005–2014). This might be due to a slight positive trend, as seen in the SIE (Fig. 3d).

In ASO (winter), the Antarctic first and second clusters show opposite-sign poles in the Weddell, Bellinghausen, and Amundsen seas, with smaller contributions from other seas (Fig. 6). These two modes resemble SIC variability driven by

Rossby wave activity across the Drake Passage described by previous literature [e.g., Yuan and Li, 2008; Hobbs and Raphael, 2010; Renwick et al., 2012; Kohyama and Hartmann, 2016]. In fact, the first cluster resembles the pattern of Antarctic SIC response to an El Niño [e.g., Ding et al., 2011] and dominates in years of strong ones, such as 1984, 1998, and 2010. The third cluster shows negative SIC anomalies along all the Antarctic perimeter but in the Bellingshausen and Amundsen seas, where anomalies are positive; this is however the least persistent cluster (11%), and SIC variability is

clearly dominated by the first two (47% and 42% respectively). Cluster occurrences and patterns in NSIDC are slightly different from those in OSI SAF and HadISTT (Supp. Figs. 2 and 3), suggesting that observational uncertainty can impact the dominant Antarctic SIC modes of variability.





### 3.3.1 Impact of ITD discretization on the SIC clusters

For each cluster of SIC variability, observations and simulations are compared mainly through their spatial correlation
(Fig. 7). As a measure of the observational uncertainty, we also calculate spatial correlation coefficients between the three
observational datasets. We further calculate the root mean square error (RMSE) across observed and simulated clusters to
provide an additional assessment. Results of the RMSE analysis are shown in the Arctic only (Supp. Fig. 4) and are
commented when they complement or disagree with results from the spatial correlation coefficients.

In the Arctic winter, correlation coefficients between observed and simulated clusters slightly decrease as the number of
categories increases in all three configurations (Fig. 7); by contrast, including more categories slightly reduces the RMSE
(which suggests a slightly better agreement with the observations) in the third cluster in the S1 and S3 configurations and
increases it in the S2 one (Supp. Fig. 4). Overall, nonetheless, the ITD configurations have a small impact on the model–data
agreement and no configuration or number of categories appear to be consistently the best.

In the Arctic summer, spread in model–data agreement is much larger than in winter (Fig. 7 and Supp. Fig. 4). The RMSE
is barely impacted by the ITD configuration (Supp. Fig. 4) and shows similar changes to those in the correlation coefficients.
The lowest model–data correlation coefficients are for the second cluster across all configurations. This is likely because of
its characteristic spatial pattern of small, mostly statistically non-significant anomalies (Fig. 5). Such noisy features are
indeed difficult to be captured by the model accurately, thus resulting in comparatively small spatial correlation coefficients.
By contrast, anomalies in the first and second clusters take larger values over a larger area and are successfully reproduced
by the simulations. Model–data spatial correlation coefficients are little influenced by the ITD configuration for the first and
third clusters but decrease with a large number of thin ice categories for the second cluster in the S1 and S3 configurations.
Although increasing the number of thick categories in the S2 configuration has no major impact on model–data correlation
coefficients, the S2.07 case shows a drop in correlation values in all the clusters. This suggests that variability is slightly
differently distributed across the clusters in this configuration. The configuration with one single category, S1.01, shows the
lowest correlation coefficients (Fig. 7) and highest RMSE values (Supp. Fig. 4) overall. These results suggest that an ITD
with one category or a large number of thin categories can potentially hamper representation of SIC variability in the Arctic.
This contrasts with and complements results in Massonnet et al. [2019], where the one-category configuration was found
performing as good as or even better than multi-category configurations in terms of sea ice mean climatology. Comparison
of mass budget across configurations showed that this configuration compensates basal ice growth deficit (relative to multi-
category cases) through a larger dynamic ice production from fall to winter (and, thus, potentially right for the wrong
reasons) [Massonnet et al., 2019].

In the Antarctica summer, model–data agreement is lower than in the Arctic in terms of both the spatial correlation (Fig.
8) and RMSE (not shown). Almost all the correlation coefficients are statistically non-significant for the second and third
cluster (Fig. 8), with only some ITD configurations with 3 or 5 categories showing significant correlations for all clusters.
For the first cluster, however, more than 5 categories seem to improve the agreement with the observations, in particular for
the S1 configuration.





In the Antarctic winter, model–data agreement increases with respect to summer, especially and correlation coefficients tend to be statistically significant for the first and especially the second cluster (Fig. 8). However, the impact of the ITD distribution is small and there is no robust response to any configuration.

### 3.3.2 Arctic SIC clusters after detrending

For a sound characterization of the modes of interannual variability over the period 1979–2014, the long-term, accelerating melting trend in the Arctic SIC is now filtered out. This trend is well captured by both the SIE (Fig. 3) and clusters (Fig. 5). Arctic SIC clusters are now calculated after detrending by removing a spatially varying 2nd degree polynomial fit with respect to time (Fig. 9). Clusters calculated after detrending with a 1st degree polynomial (linear detrend) are still affected by the melting trend and are not discussed here further. We do not consider higher order degree polynomials either, since they have shown no improvement to characterize clusters of sea ice thickness over the period 1958–2013 [Fučkar et al., 2015]. No similar analysis has been performed for Antarctic SIC as the clusters suggest a rather weak positive trend in summer (Fig. 6).

In OSI SAF, detrended SIC variability in winter is evenly distributed into the three clusters (Fig. 9; 36%, 33%, and 31% of occurrence frequency). The first cluster shows a dominant pole of negative anomalies in the Labrador Sea (Fig. 9). The second cluster shows two poles of variability of positive and negative anomalies in the Labrador and Nordic seas respectively. This cluster is very similar to the third one in not detrended data, and both dominate in similar years, in particular during periods of positive NAO phases. This suggests that they capture the fingerprint of a positive winter NAO on the Arctic SIC. The third cluster shows a clear quadrupole structure, like the first cluster in not detrended data, and dominates in similar years. Clusters in HadISST and NSIDC are very similar and shown in Supp. Figs. 4 and 5 respectively.

In summer, detrending the data leads to clusters with more marked regional contrasts (compare Figs. 5 and 9). The first cluster in OSI SAF, which dominates in two thirds of the years, shows a dipole of positive SIC anomalies in the Kara, Barents, and Greenland seas and negative ones in the East Siberian and Laptev seas (Fig. 9). The second cluster mirrors the first one but with opposite-sign and larger anomalies (Fig. 9). These two clusters respectively resemble the fingerprint of a positive (in 1995, 1999, 2002, and 2005) and negative (in 1996, and 2004) Arctic dipole on the summer SIC [Wang et al., 2009]. Occurrence of these two clusters, however, does not systematically coincide with strong Arctic dipole anomalies (for example, in 1998 or 2003; Wang et al., 2009). The Arctic oscillation has also been proposed as a driver of similar SIC anomaly patterns [Rigor et al., 2002; Rigor and Wallace, 2004; Wang et al., 2009]. Lastly, the third cluster shows a monopole of strong negative anomalies confined to the Beaufort gyre. This pattern dominates only in 4 years such as 2007, when the Arctic sea ice extent was the lowest over the period 1979–2014. Such extreme melting events have been associated with an exceptional episode of atmospheric [Graversen et al., 2010] and oceanic [Woodgate et al., 2010] warm flow into polar latitudes and summer storm activity [Screen et al., 2011]. Note that cluster repartition detrended data is not exactly the same as in HadISST and NSIDC in summer (Supp. Fig. 5): their first ones are similar to the first one in OSI SAF but with a



different local expressions; their thirds clusters resemble the second one in OSI SAF but with weaker anomalies near the
Alaskan coast.

Regarding their sensitivity to the ITD configuration after detrending, the winter clusters show a rather consistent model–
data agreement both in terms of the spatial correlation coefficients (Fig. 10) and RMSE (not shown). In summer, increasing
the number of categories beyond 30 improves the model–data correlation coefficients (Fig. 10) and reduces the RMSE (not
shown) for all the clusters in the S1 and S3 configurations (while no robust response is found in the S2 configuration). This
implies that, overall, a large number of thin categories can help improve representation of SIC interannual variability in
summer. In contrast to what happens with not detrended data, the one-category configuration agrees with the observations as
well as any other configurations, suggesting that this configuration poorly captures the forced variability but as well as any
other one the interannual variability.

### 3.4 Anomaly-based analyses

Two extra analyses are discussed in the following to complement previous ones and explore their robustness. In the first
analysis, spatial correlation coefficients are computed directly, in each year, between the simulated and observed SIC
anomalies in both seasons and hemispheres. In each case, a distribution of correlation coefficients is generated by combining
the values in all the years and in the three observational products. This analysis suggests marginal sensitivity to the number
of sea ice categories or its configuration in the Arctic before (Supp. Fig. 6) and after detrending (not shown) and in
Antarctica (not shown).

The second analysis is to provide a spatial perspective to the impacts of the ITD configurations on SIC. For this, temporal
correlation coefficients at the grid point level are first computed between simulated and observed SIC anomalies in both
seasons. The trend in such correlation coefficients with respect to the number of categories is then calculated across
simulations of a given configuration. The result is a map which provides a measure of the regions where changing the
number of categories most impacts agreement with observations. Since results are similar across the observational products,
an average between the three cases is computed for the Arctic (Fig. 11) and Antarctica (Supp. Fig. 7).

Increasing the number of categories tends to decrease model–data agreement (blue colors in Fig. 11) in the S1 and S3
configurations in both seasons (but most clearly in the S3 one in summer) in the Central Arctic, near the region where the
largest increase in sea ice thickness is simulated for an increase in the number of categories [Massonnet et al., 2019]. In that
region, a higher sea ice volume due to enhanced bottom growth rate results in a less realistic simulated sea ice. In the S2
configuration, model–data agreement particularly improves with the number of categories in winter north off Greenland and
the Queen Elizabeth Islands, regions where the thickest ice is simulated (Fig. 11, contours). Although the overall Arctic sea
ice volume increases with the number of categories [Massonnet et al., 2019], the improvement in that particular region
suggests that more categories help capture variability in thick ice. In summer, a decrease in model–data agreement occurs in
the same region, although there are improvements elsewhere in the Arctic that can potentially compensate for this decrease
(Fig. 11). In Antarctica, only the S2 configuration in summer (JFM) shows some clear trends in model–data agreement near





the Ross Sea (Supp. Fig. 7). However, these results appear spurious as sea ice is very thin and presents a concentration below 15% in the area (contours in Supp. Fig. 7).

**4 Discussion and conclusions**

This article explores the impact of different ITD configurations on the simulated SIC variability in the Arctic and Antarctica. Using ocean stand-alone simulations with the NEMO3.6–LIM3 model, we assess three different ITD configurations in which both the number and boundaries of the sea ice thickness categories are changed. SIC variability is characterized via K-means clustering analysis over the period 1979–2014; the simulated clusters are compared with those from three satellite observational products, OSI SAF (OSI-409/OSI-409-a), NSIDC (0051), and HadISST v2.2. We focus on

two seasons, JFM (winter) and ASO (summer), across which monthly clusters are most spatially coherent. In the Arctic, cluster comparison is done both including and excluding long-term trends, this latter by detrending with a spatially varying 2nd degree polynomial.

Overall, winter clusters reflect the imprint of atmospheric variability such as NAO and Siberian High on the Arctic SIC and of ENSO on the Antarctic SIC. Summer clusters reflect the dominant trends in SIC, slightly positive in Antarctica and

prominently negative in the Arctic. After detrending, Arctic summer clusters allow isolating the SIC response to atmospheric variability associated with the Arctic Dipole and Arctic Oscillation as well as identifying outstanding events such as the 2007 minimum.

Comparison between simulated and observed clusters indicates that no particular ITD configuration and number of categories systematically helps to improve the representation of model SIC variability at both poles in winter, both before

and after detrending. In summer, more thin-ice categories decrease model–data agreement at both poles before detrending, due to a poorer representation of the long-term trends; more categories, however, do improve model–data agreement in the Arctic after detrending. Nonetheless, such an improvement is found for more than 30 categories, for which computational costs substantially increases (from 30 to 60 minutes per simulated year from the standard 5-category case to 30-category one; Massonnet et al., 2019). The one-category configuration tends to show the worst results overall, particularly in the Arctic

summer before detrending. This reinforces the recommendation of using multi-category sea-ice models, such as LIM3.

Direct comparison of the SIC anomaly fields between observations and simulations suggests that increasing the number of thick categories can improve the representation of the very thick ice variability north of Greenland in winter. By contrast, including more thin categories can reduce model–data agreement in summer in the central Arctic, related to an overly large sea ice volume in the area [Massonnet et al., 2019].

Finally, comparison of SIE in simulations and observations suggests that a finer resolution of the thin ice and not of the thick ice increases realism of the simulated Antarctic summer SIE in our model.

Although the results of all these comparisons present mixed conclusions, depending on the analysis used, we can extract a few take-home messages. First, better resolving the thin ice in the Arctic can hamper SIC representation in the model, potentially related to an unrealistic sea ice volume increase, although it can improve its representation in Antarctica. Second,





more thick categories can improve the very thick ice variability in winter in the Arctic, without noticeably compromising the performance in other regions or seasons. Thus, although no clear conclusion is drawn about an optimal number of sea ice categories, our analysis does establish that configurations with more than 10 sea ice categories can degrade the realism of the simulated Arctic SIC variability. This appears counter-intuitive, as a finer resolution will allow the sea ice model to reproduce actual sea ice conditions better. Note, however, that NEMO3.6–LIM3 uses parametrizations and parameter values

that are developed to reproduce actual sea ice conditions for a 5-category configuration. Changes in the ITD configuration may therefore need re-tuning those parametrizations and parameter values. This is however beyond our scope, as improvements in model SIC variability would hence reflect the new model configurations and not solely the use of a different ITD configuration. In light of our results, we recommend using the standard (S1.05) or similar configuration in NEMO3.6–LIM3, which is, in addition, computationally more efficient.

Our study and its companion, Massonnet et al. [2019], present an advance with respect to previous efforts, since they jointly address the response of the mean climatological state and variability of the sea ice to a model parametrization. The two studies use ocean stand-alone simulations in their analysis, as to to reduce potential sources of uncertainty in SIC variability given by stochastic atmospheric noise, which might mask comparison with observations and the search for improvements in model realism. An open question for future studies is therefore whether our conclusions would hold in

coupled model configurations, where ice-atmosphere feedbacks may play a role in modulating the impact of the different ITD configurations. Despite the potential caveats, our joint approach can set an example for future assessments of the impact of model parametrizations on the representation of the sea ice or other climatic variables. Unfortunately, observational data are still too short for many climate components, and this sort of analysis is therefore challenging at most.

**Acknowledgements.** This study was funded by the European Commission's Horizon 2020 projects APPLICATE (GA 727862) and PRIMAVERA (GA 641727). Computational resources have been provided by the supercomputing facilities of the Université catholique de Louvain 5 (CISM/UCL) and the Consortium des Equipements de Calcul Intensif en Fédération Wallonie Bruxelles (CECI) funded by the F.R.S.-FNRS under convention 2.5020.11. F. Massonnet is a F.R.S.-FNRS Research Associate.


**Code/Data availability.** The model source code is available at https://doi.org/10.5281/zenodo.3345604. Data and code to reproduce the authors' work can be obtained from https://doi.org/10.5281/zenodo.3540756.

**Author contributions.** EMC and PO conceived the study. FM provided the model data. EMC analysed the model and

observational data and wrote the manuscript with contributions from all authors. All authors contributed to interpreting the results.

**Competing interests.** The authors declare that they have no competing interests.



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



**Fig. 1:** Boundaries of the ice thickness category in the three sets of sensitivity experiments. The last category's upper boundary is always set to 99 m. The thickness scale is different in the three panels. Because the third ITD configuration, S3, branches from the experiment S2.09, we repeat the latter on the bottom panel but renamed as S3.09. Figure is adapted from Massonnet et al. [2019].



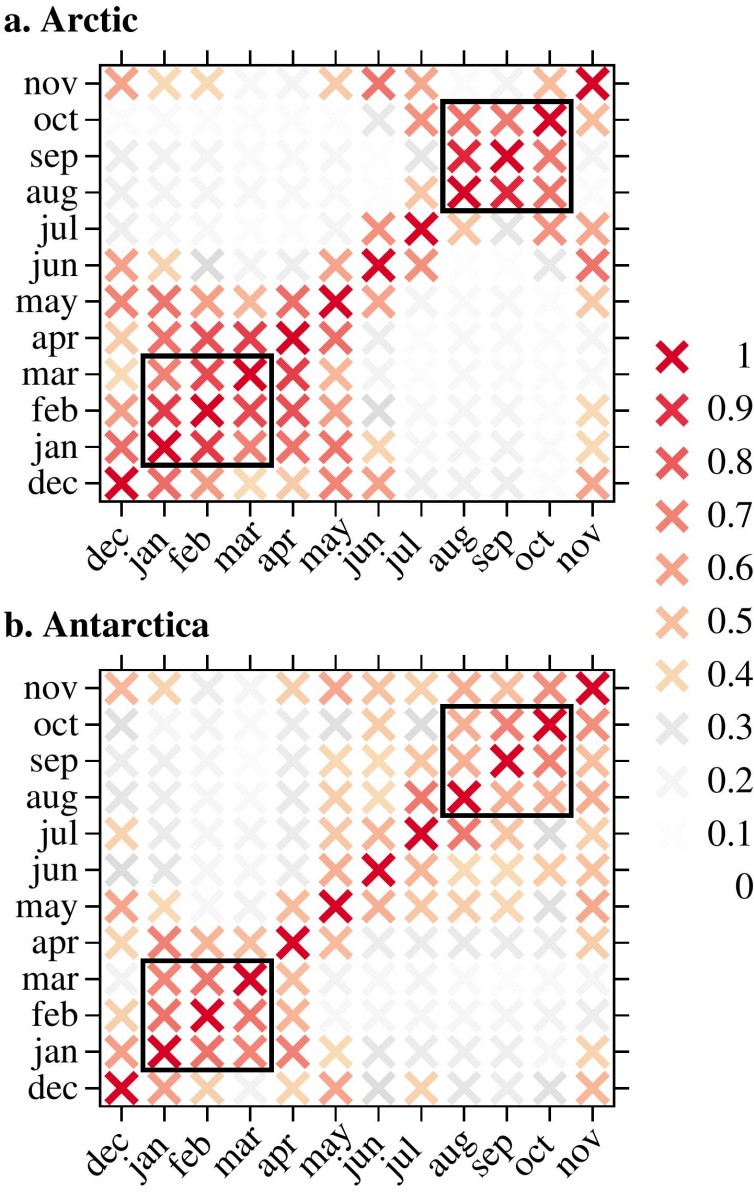

**Fig. 2:** Maximum correlation coefficient across all the monthly clusters in **(a)** the Arctic and **(b)** Antarctica in OSISAF. Two 3-month periods (seasons) stand out with the largest coefficients: January through March (JFM) and August through October (ASO). Values smaller than 1/e (~0.37) are considered statistically non-significant and plotted in gray. Similar results are obtained using NSIDC and HadISST SIC (and therefore not shown).





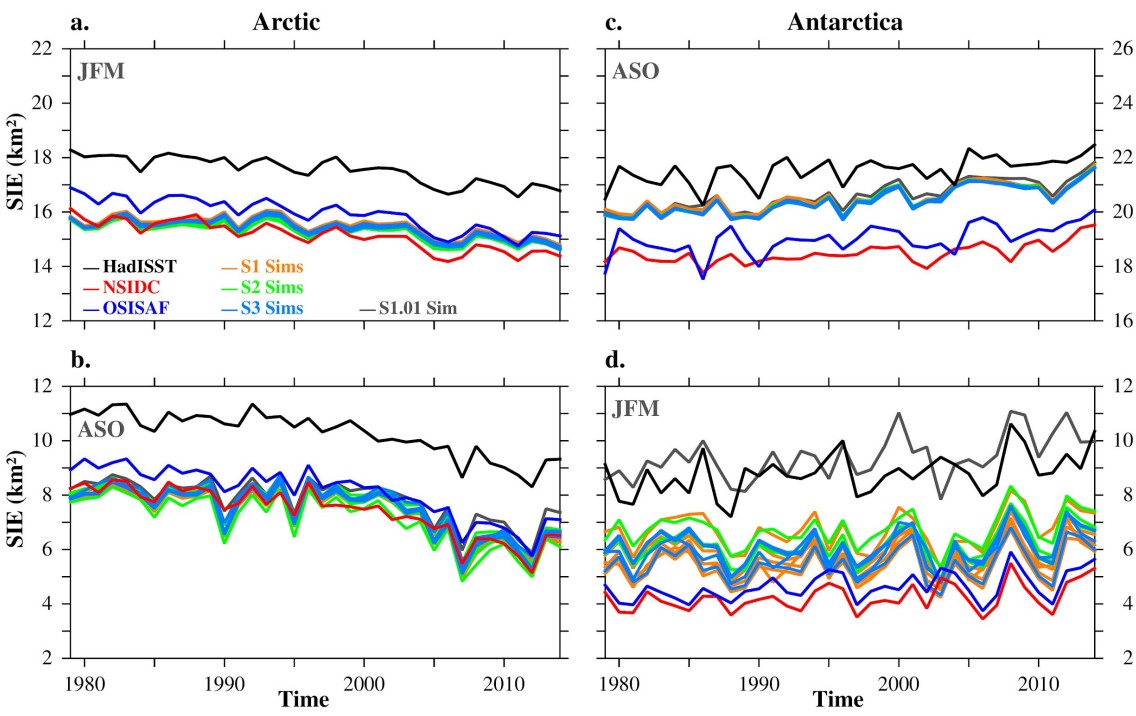

**Fig. 3: (a,b)** Arctic and **(c,d)** Antarctic sea ice extent (SIE; in km²) in **(a,d)** JFM and **(b,c)** ASO in the S1, S2, and S3 configurations (in light red, green, and blue respectively, with the configuration with one single category, S1.01, in gray) and in HadISST, NSIDC, and OSI SAF (in black, blue, and red respectively). SIE is calculated as the total area of grid cells with concentration larger than 15 %.





**Fig. 4:** Integrated ice edge error (IIEE; in km²) between the Antarctic sea ice in JFM in the S1 (top; with the case with one single category, S1.01, in gray), S2 (middle), and S3 ITD configurations (bottom) and in HadISST (dotted lines), NSIDC (dashed lines), and OSISAF (solid lines). Also, IIEE is shown between OSI SAF and NSDIC SIC (gray crosses), NSIDC and HadISST SIC (gray pluses), and HadISST and OSI SAF SIC (gray asterisks). The IIEE is calculated as the integrated area where simulations and observations disagree on SIC above 15% [Goessling et al., 2016]. The darker the color of the line is, the more categories that ITD configuration has. The color scheme matches that in Fig. 1

**Fig. 5:** Caption next page.





**Fig. 5:** (Previous page). *First three rows:* cluster patterns of Arctic SIC anomalies (shading; in % of area) in OSI SAF in JFM (left) and ASO (right). Stippling masks statistically non-significant anomalies at the 5% level; p-values at each grid point are computed through a *t*-test that accounts for serial autocorrelation [Manubens et al., 2018]. Each cluster's percentage occurrence over the period 1979–2014 is indicated in each case. The shading color scale is adapted for a better view of the anomalies in the range ±15%. The area is zoomed in in ASO (right) for a better view of the central Arctic. *Fourth row:* time series of cluster occurrence in HadISST (black crosses), NSIDC (red diamonds), and OSISAF (blue pluses). The larger the symbol size, the larger the Euclidean distance (root mean square difference) between a pattern of anomalies and the associated cluster in a particular year (the maximum symbol size is shown in the legend). Clusters are calculated from the full SIC field without detrending (in contrast to detrended data shown in Fig. 9)







**Fig. 6:** As in Fig. 5 but in Antarctica.





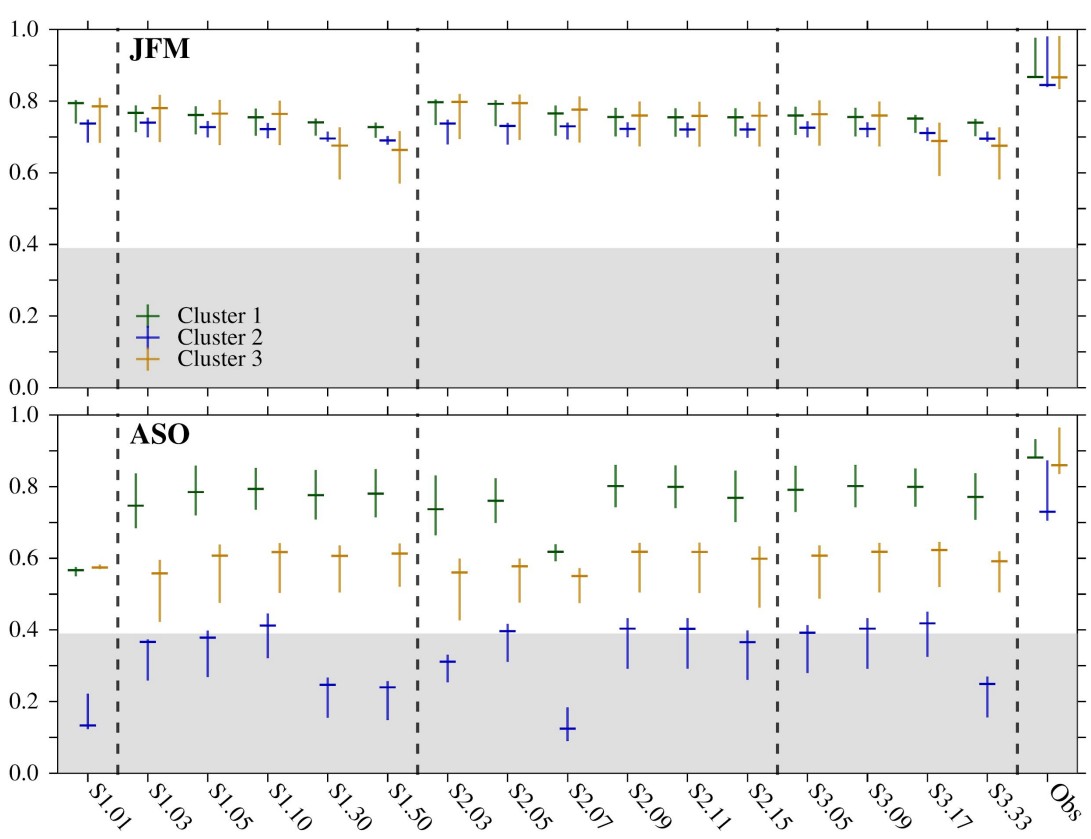

**Fig. 7:** Spatial correlation coefficients between the simulated and observed clusters and across the three satellite observational products (marked as Obs) of Arctic SIC in JFM (top) and ASO (bottom). For each case, the vertical line spans the maximum and minimum correlation coefficients, and the horizontal line marks the middle one; green, blue, and orange lines are for the first, second, and third clusters. Gray shading masks statistically non-significant coefficients below 0.39 value, which corresponds with the minimum value across all the computations that is statistically significant at the 5% level, accounting for effective degrees of freedom and spatial autocorrelation. Dashed vertical lines separate between results in the simulation with one single category (S1.01), the different ITD configurations (S1, S2, and S3), and the observations. Note the configuration S2.09 and S3.09 are the same (Fig. 1).





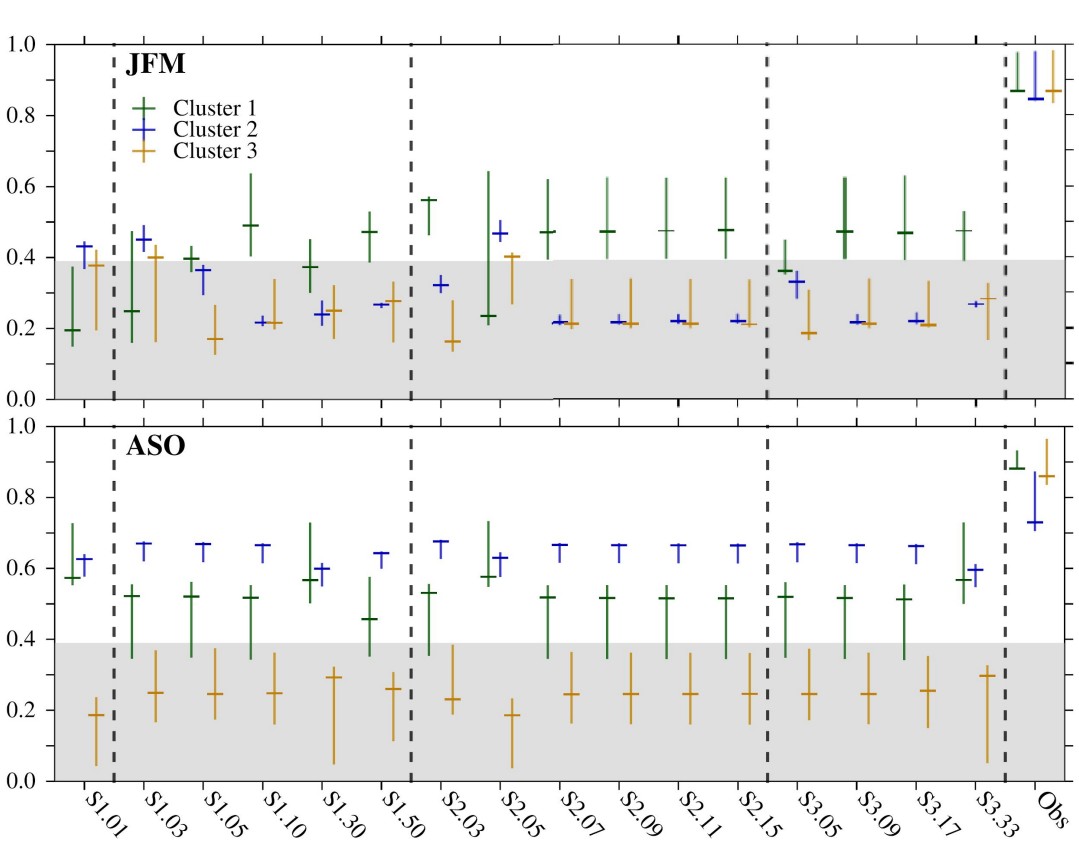

**Fig. 8:** As in Fig. 7 but for clusters in Antarctica.



**Fig. 9:** As in Fig. 5 but after detrending with a 2nd degree polynomial.





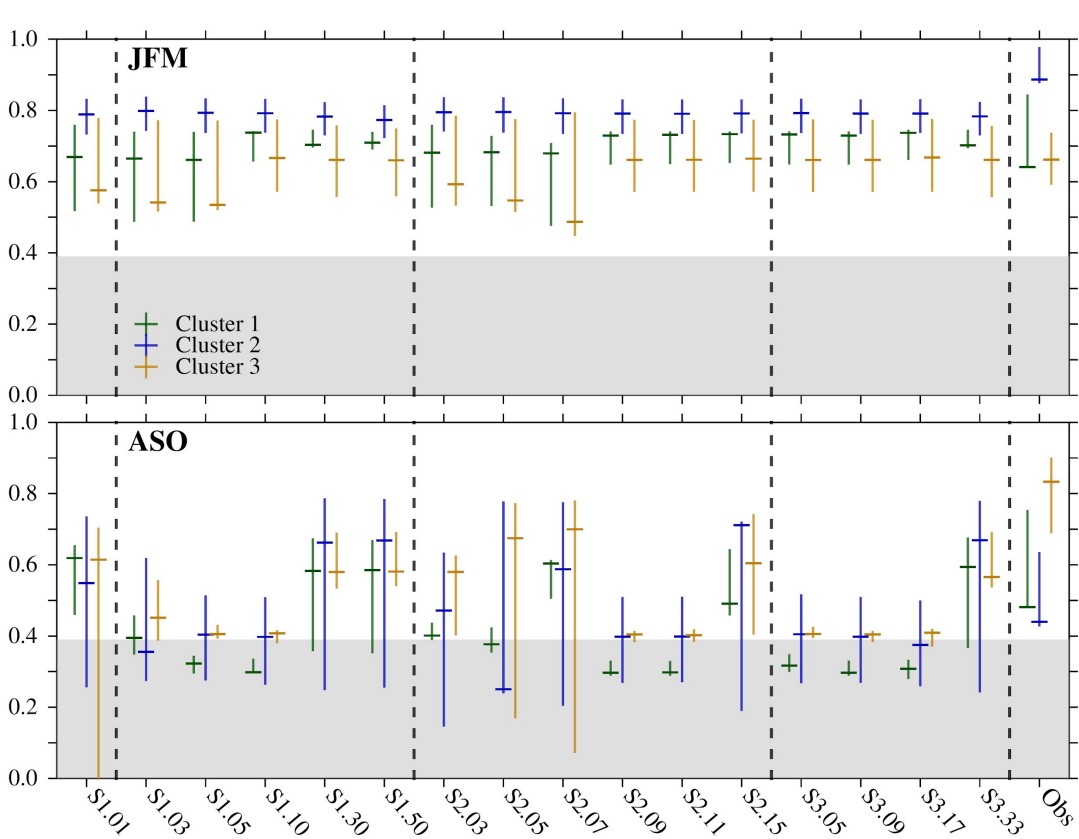

**Fig. 10:** As in Fig. 7 but after detrending with a 2nd degree polynomial.





**Fig. 11:** Caption next page.



**Fig. 11:** *(Previous page)* Trend relative to the number of sea ice categories in pointwise temporal correlation coefficients between simulated and observed Arctic SIC anomalies (in [*number of categories]⁻¹*) in the S1 (top), S2 (middle), and S3 (bottom) configurations in JFM (left) and ASO (right). An average between results in OSI SAF, NSIDC, and HadISST is shown. Values are multiplied by 100 to ease the interpretation and the color shading is adapted for a better view of the values between –1 and 1. A value of 0.5, for example, indicates that the model–data correlation coefficient increases 0.5 when the number of categories increases by 100 (or 0.25 for an increase of 50 categories). Stippling masks trend values that statistically significant at the 5 % level based on a two-tailed Student's *t* test. Contours are the simulated climatological ice thickness (every 1.5 m) in the standard LIM3 configuration of 5 categories (S1.05 configuration) for the period 1979–2014.