# Peer review of "Impact of the ice thickness distribution discretization on the sea ice concentration variability in the NEMO3.6-LIM3 global ocean—sea ice model"

_Geoscientific Model Development, 2019_

## Referee Comment (RC1) · Anonymous Referee #1 · 23 Dec 2019

This is a well written manuscript addressing the topic of sea-ice thickness distribution configuration, which is important for those who use the NEMO-LIM ocean-sea-ice model. Therefore the paper is suitable to be published in the Nucleus for European Modelling of the Ocean - NEMO special issue. The manuscript provides useful results for NEMO-LIM modellers based on advanced statistical and visualisation methods that appear valid. To my opinion, these results sufficiently substantially advance in modelling science. In particular, the main result that no clear benefit is obtained from increasing the number of sea ice thickness categories beyond the current usual standard

of 5 categories in NEMO3.6-LIM3 is useful so that a model user will not to waste time in testing a range of number of sea-ice categories for better results.

In Discussion the authors conclude that changes in the ice thickness distribution configuration need re-tuning parametrizations and parameter values. Would be useful for future approaches to list which parameters needed re-tuning. In the current version no specifics has been discussed.

The other aspects for a reviewer to consider, seem adequately addressed too, but there are a few things that might be useful for the NEMO community if added or expanded the paper. There are also a small amount of corrections that the text requires. Due to these, a minor revision is required with the details following.

Detailed comments:

- line 14 'coherence across' would be more precise to say 'correlation across'

- line 16 Here 'atmospheric variability' does not point to synoptic one, as one might guess when reading the abstract, but longer, large-scale atmospheric modes. This could be specified by rewriting 'long-term atmospheric variability'.

- line 24 You could mention why there is 'overly large simulated sea-ice growth'. Is it due to the fact that thin ice grows faster?

- line 26 'Antarctica' comes sudden here as improvements there has not been mentioned before in the abstract. I suggest adding a sentence how Antarctic sea-ice was improved by better resolved thin ice after the sentence ending in line 21.

- line 33. '... Antarctica. These modes drive ...'

- line 40. '... variability in modes such as the NAO ...'

- line 45. '...determines its important physical processes, such as salt and ...'

- line 81. To me 167 mm/day is not weak but strong restoring. Drop word 'weak' in line 80.

- line 81. 'concentratio' -> 'concentration'

- line 127. '... (namely Duda-Hart ...'

- line 142. '... the optimal number ...'

- line 153. '... clusters presented later ...'

- line 171. '... emerging from .'

- line 244. '... configuration with single category ...'

- line 247. '... where the single category ...'

- line 252. 'In the Antarctic summer ...'

- line 257. '... increases especially with respect ...'

- line 287. '... repartition of detrended data ...'

- line 289. '... their third clusters ...'

- line 294. '... all the clusters in the S3 configurations ...'. In S2 max categories is 15, so it says nothing can be said about categories beyond 30.

- line 298. '... other configuration, the ...'

- line 303. '... suggests only marginal ...'

[Figure]

- line 308. Using the word 'trend' in this context is confusing because trend is commonly understood as a change in time. I suggest you replace 'trend' with e.g. linear fit or something else more suitable.

- line 315. Is enhanced bottom grow because thin ice grows faster? You should explain the physics behind the enhanced bottom grow.

- line 319. '... more thick categories ...'

- line 320. '... in the Central Arctic that can potentially compensate for this decrease in terms of SIE ...'

- line 329. Explain what expression in parenthesis mean in OSI-SAF and NSIDC. Are they needed here? Data are already described in section 2.

- line 330. '... done by both including and excluding long-term trends.'

- line 336. '... such as the 2007 SIE minimum'.

- line 350. This is a one-sentence paragraph. Merge it with the earlier one.

- line 360. '... are adjusted to reproduce ...'

- line 364. '... computationally more efficient than configurations with more categories.'

- line 366. '... sea ice to changes in model parametrization.'

- Fig 5. caption. 'anomalies in the range of $\pm 15$

---

## Referee Comment (RC2) · Anonymous Referee #2 · 6 May 2020

**Moreno-Chamarro et al.: Impact of the ice thickness distribution discretization on the sea ice concentration variability in the NEMO3.6-LIM3 global ocean–sea ice model**

The authors investigate ice thickness distribution (ITD) categories in NEMO-LIM and how they impact sea ice concentration variability. They use k-means clustering as a technique in tandem with three observational based SIC datasets. The authors do not find an optimal configuration as results in the Arctic and Antarctic have opposite responses to ITD changes, so no clear benefit to NEMO-LIM is determined from changing ITD.

Overall, I believe this will be suitable for publication with a few major/moderate changes. I felt that the scientific significance and quality were good to fair, but could be improved with some expansion in the text. The Scientific Reproducibility is also fair, which again could be improved with further clarification in the text. The Presentation quality was excellent.

**Specific Comments:**

- One of the biggest concerns I have about this paper is that it doesn't generalize to modeling in general beyond NEMO-LIM to provide insight about modeling in general. I realize that this is for the NEMO special issue, however, it currently feels a bit like a sensitivity experiment to determine optimal model configuration but not otherwise generally of interest to the community of sea ice modelers who may be setting up their own models using LIM or other sea ice models.

  This begins in the introduction where there should be a brief discussion of previous work about why 5 ITD categories have been chosen in the past due to volume studies (Lipscomb 2001, Remapping the thickness distribution in sea ice models, doi: 10.1029/2000JC000518; Bitz et al. 2001, Simulating the ice-thickness distribution in a coupled climate model, doi: 10.1029/1999JC000113). In fact, in Bitz 2001 one of the conclusions is "…the concentration of open water and thin ice, which is relatively insensitive to the number of categories beyond M=5," which is directly relevant for this paper. Why weren't these cited? If anything, studies using CICE that agree with these results should strengthen your results because they become more robust across models.

  In the discussion and conclusions section you should add more information about how these results might be directly relevant in coupled models. This is brought up briefly but could be fleshed out and suggestions for how to test this would be useful. Additionally, you mention that parameterizations and parameter values are tuned for 5 categories (line 359). Can you specify which of these might be directly affected or changed? Are similar parameterizations present in other sea ice models? How can this be generalized for the community?

- The methods need clarification, particularly for replicability purposes. In particular, I found these sections to need to be expanded. 1) At line 145/Figure 2 the Arctic "winter" cluster was defined but didn't include April. What threshold values for these groups were used? Are you results sensitive to including different months? These things should be tested. 2) The % values in Fig.5/6 refer to occurrence, can you translate these values to number of months or something to better indicate what this means? 3) Section 3.3.1 – are these correlation differences statistically significant from one another? Can you clarify what you mean by these are significant? 4) Line 263 – how was the polynomial determined? Can you provide information about this?

- If there is not a lot of information gleaned from the de-trended Arctic analysis, then why is it presented? Can this be condensed somehow since the variability analysis primarily shows the forced trend without being de-trended?

- I think that if possible you should consider including Supplemental Figures 4 and 7 as regular figures since they are referred to in detail.

**Technical corrections:**
- Line 81: misspelled "concentration"

- Lines 269-275: It looks to me like patterns 2 and 3 are both dipoles but opposite patterns. Can you clarify where the quadrupole is?

- Lines 296-298: sentence is confusing. "…suggesting that this configuration poorly captures the forced variability but does capture interannual variability as well as any other configuration."?

- The stippling markers are used to indicate significance in Fig. 11 but insignificance in Fig.5. It would be nice if they were used consistently.

- The first two paragraphs of the discussion were clear and concise. The next three are a bit confusing and all over the place. I'd suggest you rearrange in the following order: 1. One category has worst results necessitating multi-category sea ice models like LIM3 or CICE; 2. The standard configuration is 5 ITD levels; 3. Adding more thin categories decreases agreement; 4. Having 30+ categories can improve some but is significantly more expensive at double the cost, which is clearly significant for coupled models.

---

## Author Response (AR1)

**Reviewer 1**

This is a well written manuscript addressing the topic of sea-ice thickness distribution configuration, which is important for those who use the NEMO-LIM ocean-sea-ice model. Therefore the paper is suitable to be published in the Nucleus for European Modelling of the Ocean - NEMO special issue. The manuscript provides useful results for NEMO-LIM modellers based on advanced statistical and visualisation methods that appear valid. To my opinion, these results sufficiently substantially advance in modelling science. In particular, the main result that no clear benefit is obtained from increasing the number of sea ice thickness categories beyond the current usual standard of 5 categories in NEMO3.6-LIM3 is useful so that a model user will not waste time in testing a range of number of sea-ice categories for better results.

We thank the Reviewer for the appreciation and the thoughtful comments. In the following we answer each specific point (in blue).

In Discussion the authors conclude that changes in the ice thickness distribution configuration need re-tuning parametrizations and parameter values. Would be useful for future approaches to list which parameters needed re-tuning. In the current version no specifics has been discussed. We referred to the typical parameters of the sea ice models, which include, among others, snow thermal conductivity, bare sea-ice albedo, and compressive ice strength, P\*. This is now discussed in the revised manuscript (Lines 383–389).

The other aspects for a reviewer to consider, seem adequately addressed too, but there are a few things that might be useful for the NEMO community if added or expanded the paper. There are also a small amount of corrections that the text requires. Due to these, a minor revision is required with the details following.

**Detailed comments:**

• line 14 'coherence across' would be more precise to say 'correlation across'

**Corrected.**

• line 16 Here 'atmospheric variability' does not point to synoptic one, as one might guess when reading the abstract, but longer, large-scale atmospheric modes. This could be specified by rewriting 'long-term atmospheric variability'.

**Clarified as suggested**

• line 24 You could mention why there is 'overly large simulated sea-ice growth'. Is it due to the fact that thin ice grows faster?

Massonnet et al. [2019] show that this is because of a net increase in basal ice growth rate, which is indeed promoted when the relative area of thin ice is large. This is indicated in the revised manuscript now (Line 23)

• line 26 'Antarctica' comes sudden here as improvements there has not been mentioned before in the abstract. I suggest adding a sentence how Antarctic sea-ice was improved by better resolved thin ice after the sentence ending in line 21. Done.

• line 33. '... Antarctica. These modes drive ...' The sentence has been clarified.

• line 40. '... variability in modes such as the NAO ...' Corrected.

• line 45. '...determines its important physical processes, such as salt and ...'

We think the word properties, as in the original manuscript, is more precise than processes to describe quantities like heat capacity or resistance to deformation. The line therefore has not been modified.

• line 81. To me 167 mm/day is not weak but strong restoring. Drop word 'weak' in line 80. Done.

line 81. 'concentratio' -> 'concentration'
 Corrected.

• line 127. '... (namely Duda-Hart ...' Corrected. • line 142. '... the optimal number ...' Corrected.

• line 153. '... clusters presented later ...' Corrected.

• line 171. '... emerging from ..' Corrected.

• line 244. '... configuration with single category ...' Corrected.

• line 247. '... where the single category ...' Corrected.

• line 252. 'In the Antarctic summer ...' Corrected.

• line 257. '... increases especially with respect ...' Corrected.

• line 287. '... repartition of detrended data ...' Corrected.

• line 289. '... their third clusters ...' Corrected.

line 294. '... all the clusters in the S3 configurations ...'. In S2 max categories is 15, so it says nothing can be said about categories beyond 30.
 Corrected.

• line 298. '... other configuration, the ...' Corrected.

• line 303. '... suggests only marginal ...' Added.

• line 308. Using the word 'trend' in this context is confusing because trend is commonly understood as a change in time. I suggest you replace 'trend' with e.g. linear fit or something else more suitable.

Corrected as suggested.

• line 315. Is enhanced bottom grow because thin ice grows faster? You should explain the physics behind the enhanced bottom grow.

This has been explained (Lines 339–342).

• line 319. '... more thick categories ...' Added.

• line 320. '... in the Central Arctic that can potentially compensate for this decrease in terms of SIE ...'

Added.

• line 329. Explain what expression in parenthesis mean in OSI-SAF and NSIDC. Are they needed here? Data are already described in section 2.

This has been removed as that information is already given in Section 2.

• line 330. '... done by both including and excluding long-term trends.' Corrected.

• line 336. '... such as the 2007 SIE minimum'. Added.

• line 350. This is a one-sentence paragraph. Merge it with the earlier one. All the Discussion section has been reformulated. • line 360. '... are adjusted to reproduce ...' Modified as suggested.

• line 364. '... computationally more efficient than configurations with more categories.' Added.

• line 366. '... sea ice to changes in model parametrization. Added.

• Fig 5. caption. 'anomalies in the range of ±15 Corrected.

**Reviewer 2**

Moreno-Chamarro et al.: Impact of the ice thickness distribution discretization on the sea ice concentration variability in the NEMO3.6-LIM3 global ocean–sea ice model. The authors investigate ice thickness distribution (ITD) categories in NEMO-LIM and how they impact sea ice concentration variability. They use k-means clustering as a technique in tandem with three observational based SIC datasets. The authors do not find an optimal configuration as results in the Arctic and Antarctic have opposite responses to ITD changes, so no clear benefit to NEMO-LIM is determined from changing ITD.

Overall, I believe this will be suitable for publication with a few major/moderate changes. I felt that the scientific significance and quality were good to fair, but could be improved with some expansion in the text. The Scientific Reproducibility is also fair, which again could be improved with further clarification in the text. The Presentation quality was excellent.

We thank the Reviewer for the appreciation and the thoughtful comments. In the following we answer each specific point (in blue).

Specific Comments:

• One of the biggest concerns I have about this paper is that it doesn't generalize to modeling in general beyond NEMO-LIM to provide insight about modeling in general. I realize that this is for the NEMO special issue, however, it currently feels a bit like a sensitivity experiment to determine optimal model configuration but not otherwise generally of interest to the community of sea ice modelers who may be setting up their own models using LIM or other sea ice models.

This begins in the introduction where there should be a brief discussion of previous work about why 5 ITD categories have been chosen in the past due to volume studies (Lipscomb 2001, Remapping the thickness distribution in sea ice models, doi: 10.1029/2000JC000518; Bitz et al. 2001, Simulating the ice-thickness distribution in a coupled climate model, doi: 10.1029/1999JC000113). In fact, in Bitz 2001 one of the conclusions is "...the concentration of open water and thin ice, which is relatively insensitive to the number of categories beyond M=5," which is directly relevant for this

paper. Why weren't these cited? If anything, studies using CICE that agree with these results should strengthen your results because they become more robust across models.

We thank the reviewer for the references. We opted for an Introduction briefly reviewing previous research on the impact of the ITD in climate models since a longer, more detailed one is provided in the companion paper Massonnet et al. (2019). This was also done because nearly all of the previous studies have focused on the mean climatological state of the sea ice (the focus in Massonnet et al., 2019) and not on its variability (our focus). Thus, whereas Bitz et al. (2001) was indeed cited in the Introduction as an example, Lipscomb (2001) was not. Both works are now cited in the revised manuscript. The Introduction has further been clarified on this point and extended following the Reviewer's suggestion (Lines 48–55).

In the discussion and conclusions section you should add more information about how these results might be directly relevant in coupled models. This is brought up briefly but could be fleshed out and suggestions for how to test this would be useful. Additionally, you mention that parameterizations and parameter values are tuned for 5 categories (line 359). Can you specify which of these might be directly affected or changed? Are similar parameterizations present in other sea ice models? How can this be generalized for the community?

The Discussion section has been rewritten in full to accommodate these suggestions. Now dedicated parts discuss the tuning parameters that might need adjustment in the model (Lines 383–389), the potential relevance for other sea ice models beyond NEMO3.6-LIM3 (Lines 390–406) and for fully-coupled modes (Lines 407–415).

• The methods need clarification, particularly for replicability purposes. In particular, I found these sections to need to be expanded.

1) At line 145/Figure 2 the Arctic "winter" cluster was defined but didn't include April. What threshold values for these groups were used? Are your results sensitive to including different months? These things should be tested.

We opted for the standard definition of a season of three months. To define the winter (summer) season, we search for the largest correlation coefficient between the monthly clusters and the adjacent second largest value in the winter (summer) half year (correlations coefficients are plotted in Fig. 2). The two seasons must be and are

consistent across the three observational datasets included in the analysis. This method renders two seasons in which monthly cluster agreement is consistently high: JFM and ASO, on which we then base the whole paper analysis. Although monthly clusters in April and in previous months show good agreement, agreement is smaller than across JFM. This method hence leaves April outside the winter season. We note our winter season agrees with the analysis in Close et al., 2017, where monthly Principal Components of the sea ice concentration show that January, February, and March have similar modes, different from those in November, December, or April. For these reasons, we have decided to keep our definition of a winter season without April. This point has nonetheless been clarified in the revised manuscript (Lines 156–160).

In addition, to show that including April in winter would actually have had little impact on the analysis, Response Figure 1 (below) shows the test where the clusters are extracted in the 4-month seasons January–April (as winter) and July–October (as summer) in the Arctic. The main difference with respect to the clusters in JFM and ASO (Fig. 5) is that the second and third clusters have switched positions. Cluster patterns and years of occurrence are however virtually identical as those in JFM and ASO (c.f. Fig. 5). The Response Figure 1 is not included in the received manuscript, but we do mention in the text that results are not sensitive to the specific season definitions.

Response Figure 1: As in Fig. 5 in the main text, but in JFMA (left) and JASO (right)

2) The % values in Fig.5/6 refer to occurrence, can you translate these values to number of months or something to better indicate what this means?

The value is the percent of years in the period 1979–2014 whose anomaly pattern is the closest to a particular cluster. The number of months is now shown together with the percent value in Figs. 5 and 6, and Supp. Figs. 2, 3, and 5.

3) Section 3.3.1 – are these correlation differences statistically significant from one another? Can you clarify what you mean by these are significant?

In the updated manuscript, statistical significance of the difference between correlation coefficients is tested using Fisher's z-transform assuming a two-tailed significance level

of 0.05. Given the large number of coefficient pairs for which differences might be tested, we simplify this by comparing only the median values between an ITD discretization and the one immediately below within the same discretization type: for example, S1.50 is compared with S1.30, the latter with S1.10, and so on; S2.15 is compared with S2.11, and so on; S1.03, S2.03, and S3.05 are all compared with S1.01. The results of these significance tests are now included and discussed in the revised manuscript (see the new Figs. 7, 8 and 10, and Supp. Fig. 6).

4) Line 263 – how was the polynomial determined? Can you provide information about this?

Detrending is done by removing a spatially varying 2nd degree polynomial fit with respect to time using the 'Trend' function in the s2dverification R package [Manubens et al., 2018]. This is now indicated in the revised manuscript (Lines 145–146).

• If there is not a lot of information gleaned from the de-trended Arctic analysis, then why is it presented? Can this be condensed somehow since the variability analysis primarily shows the forced trend without being de-trended?

We would like to keep this section in the main text. The analysis of detrended data is actually critical to characterizing interannual variability in summer, a season which is dominated by the long-term melting trend in the Arctic. Without detrending, Arctic clusters mostly capture this trend (compare Figs. 5 and 9 in the main text). The analysis of detrended data further shows that ice thickness distributions with more than 30 categories can help improve model–data agreement in the Arctic, at a cost of making the simulations computationally more expensive.

I think that if possible you should consider including Supplemental Figures 4 and 7 as regular figures since they are referred to in detail.
 We would like to keep them in the Supplement. Both Figures are only briefly discussed in the manuscript and add little extra information to the discussion of the results. And although the number of figures is not a constraint for publication in GMD, we think 11 main Figures is already a high enough number.

Technical corrections:

Line 81: misspelled "concentration" Corrected

Lines 269-275: It looks to me like patterns 2 and 3 are both dipoles but opposite patterns. Can you clarify where the quadrupole is?

We call quadrupole a cluster that shows four dominant poles in Arctic sea ice concentration, regardless of the sign. In the winter Arctic this usually means a pole in the Labrador, Barents-Greenland, Okhotsk, and Bering seas respectively. This follows the definition of the quadrupole in Close et al., 2017. Both clusters 2 and 3 in Fig. 9 would therefore fall in this definition. We note, however, small differences between the two. In cluster 2 the pole in the Labrador Sea dominates and dominates in years of strong positive winter NAO. We interpret this as the wind-driven signature of the NAO on the ice concentration [Bader et al., 2011]. Cluster 3 is instead closer to cluster 1 in not detrended data and dominates in similar years. They both further resemble the quadrupole pattern analyzed in Close et al., 2017. This point has been clarified in the revised manuscript.

Lines 296-298: sentence is confusing. "...suggesting that this configuration poorly captures the forced variability but does capture interannual variability as well as any other configuration."? This has been clarified.

The stippling markers are used to indicate significance in Fig. 11 but insignificance in Fig.5. It would be nice if they were used consistently.

Both Fig. 11 and Supp. Fig. 7 have been modified as suggested.

The first two paragraphs of the discussion were clear and concise. The next three are a bit confusing and all over the place. I'd suggest you rearrange in the following order: 1. One category has worst results necessitating multi-category sea ice models like LIM3 or CICE; 2. The standard configuration is 5 ITD levels; 3. Adding more thin categories decreases agreement; 4. Having 30+ categories can improve some but is significantly more expensive at double the cost, which is clearly significant for coupled models

Both the Discussion and Abstract have been rewritten following the Reviewer's recommendation.

**Impact of the ice thickness distribution discretization on the sea ice concentration variability in the NEMO3.6-LIM3 global ocean-sea ice model**

Eduardo Moreno-Chamarro1\*, Pablo Ortega1, and François Massonnet2

1Barcelona Supercomputing Center (BSC), Barcelona, Spain. 2Georges Lemître Centre for Earth and Climate Research, Earth and Life Institute, Université catholique de Louvain, Louvain-la-Neuve, Belgium

Correspondence to: Eduardo Moreno-Chamarro (eduardo.moreno@bsc.es)

5

20

- Abstract. This study assesses the impact of different sea ice thickness distribution (ITD) configurations\_discretizations on the

   10
   sea ice concentration (SIC) variability in ocean-standalone NEMO3.6-LIM3 simulations. Three ITD configurations\_discretizations with different numbers of sea ice thickness categories and boundaries are evaluated against three different satellite products (hereafter referred to as "data"). Typical model and data interannual SIC variability is characterized by kK-means clustering both in the Arctic and Antarctica between 1979 and 2014-in. We focus on two seasons, winter (January–March) and summer (August–October), when coherence in which correlation coefficients across clusters in
- 15 individual months is are largest. Analysis in In the Arctic is done, clusters are computed before and after detrending the series with a 2nd degree polynomial to separate interannual from longer-term variability.

Before\_The analysis shows that, before detrending, winter clusters capturereflect the SIC response to large-scale atmospheric variability at both poles and summer cluster a positive and negative trend in the, while summer clusters capture the negative and positive trends in Arctic and Antarctic SIC respectively. After detrending, Arctic clusters reflect the\_SIC response to interannual atmospheric variability predominantly. Model\_data cluster comparison suggests that no specific ITD-

- configuration or category number increases realism of the simulated Arctic and Antarctic SIC variability in winter. In the Arctic summer, more thin-ice categories decrease model-data agreement without detrending but increase agreement after detrending. Overall, a single-category configuration agrees the worst with data.
- Direct model-data comparison of SIC anomaly fields shows that more thick-ice categories improve winter SIC variability realism in Central Arctic regions with very thick-ice. By contrast, more thin-ice categories reduce model-data agreement in the Central Arctic in summer, due toThe cluster analysis is complemented with a model-data comparison of the sea ice extent and SIC anomaly patterns.

The single-category discretization shows the worst model-data agreement in the Arctic summer before detrending, related to a misrepresentation of the long-term melting trend. Similarly, increasing the number of thin categories reduces

**30 model-data agreement in the Arctic, due to a poor representation of the summer melting trend, and an overly large simulatedwinter sea ice volume-**

In summary, whereas better resolving thin ice in NEMO3.6-LIM3 can hamper model realism in the Arctic but improve it associated with a net increase in basal ice growth. In contrast, more thin categories improve model realism in Antarctica, and more thick-ice categories increase realism in the Arctic winter. And although the single-category configuration performs

35 the worst overall, no optimal configuration is identified ones improve it in Central Arctic regions with very thick ice. In all the analyses we nonetheless identify no optimal discretization. Our results thus suggest that no clear benefit in the representation of SIC variability is obtained from increasing the number of sea ice thickness categories beyond the current usual-standard of with 5 categories in NEMO3.6-LIM3. Formatted: Normal, Indent: First line: 0 cm, Line spacing: single

**40 1 Introduction**

AnalysesAnalysis of recent observations havehas allowed identifying different drivers of sea ice variability. For example, interannual sea ice variability has primarily been related to Interannual sea ice variability, for example, has been associated primarily with changes in atmospheric and oceanic circulation: atmospheric variability<del>, which can directly be</del> related to largescale atmospheric modes, such as the North Atlantic Oscillation (NAO) or Siberian High in the Northern Hemisphere, and the

- 45 Southern Annular Mode over Antarctica, can drive changes in the sea ice both dynamically and thermodynamically [e.g., Rigor et al., 2002; Rigor and Wallace, 2004; Ogi et al., 2007; Yuan and Li, 2008; Wang et al., 2009; Hobbs and Raphael, 2010; Holland and Kwok, 2011; Renwick et al., 2012; Kohyama and Hartmann, 2016; Lynch et al., 2016; Close et al., 2017; Blackport et al., 2019; Olonscheck et al., 2019]. Similarly, interannual changes in ocean heat transport to high latitude can contribute to anomalous episodes of Arctic sea ice melting in both the Atlantic and Pacific sectors [e.g., Hibler, 1986; Venegas and
- 50 Mysak, 2000; Ingvaldsen et al., 2004a; 2004b; Woodgate et al., 2010; Schlichtholz, 2011]. On longer time scales, the accelerating thinning in Arctic sea ice [Comiso et al., 2008; Serreze and Stroeve, 2015] might be modulated by lower-frequency variability in modes likesuch as the NAO [e.g., Delworth et al., 2016] or Atlantic Multidecadal Variability [e.g., Day et al., 2012; Drinkwater et al., 2014; Miles et al., 2014]. Accurately capturing this complex range of variability in sea ice, together with itsthe potential impacts on the lower latitude climate [e.g., Screen, 2013], demands for a realistic representation of the sea ice in climate models.

One among the many crucial features of sea ice to ensure its realistic representation is its thickness complexityheterogeneity, which determines other important physical properties, such as ice's salt and heat content, resistance to deformation and fracture, and melting and growth rates. State-of-the-art sea ice models typically use an ice thickness distribution (ITD) [Thorndike et al., 1975] to account for subgrid-scale variability of ice properties. In most cases,

- 60 through an ITD the different ice thicknesses are sorted into a fixed number of categories in a configuration which with usually presents the finest resolution in the thinnest ice range. Several studies have explored the advantages of including an ITD to simulate the mean state and seasonality inof the sea ice accurately, as well as the number of categories that would render theits most realistic ice representation [e.g.representation, albeit with mixed results [among others, Bitz et al., 2001; Lipscomb, 2001; Holland et al., 2006; Massonnet et al., 2011; Uotila et al., 2017; Ungermann et al., 2017; and Massonnet et
- 65 al., 2019]. These studies, however, have partly overlooked Although 5 to 7 categories were initially found sufficient to simulate large-scale sea ice realistically [Bitz et al., 2001; Lipscomb, 2001], the later study by Hunke [2014] concluded that such numbers might lead to an inaccurate representation of the observed sea ice thickness and a model misrepresentation of mechanical sea ice processes controlling its volume. The optimal number of categories and discretization are therefore still debated (a more detailed review is given in the companion paper, Massonnet et al., 2019). Interestingly, we note that
- 70 these previous studies partly overlook the impact of the ITD discretization on the simulated sea ice variability. To our knowledge, only Massonnet et al. [2011] reported a more realistic interannual variability in the Arctic sea ice extent (SIE) in the LIM3 sea-ice model than in itsthe previous model version, LIM2 (although this improvement cannot exclusively be attributed to the addition of an explicit 5-category ITD in LIM3 but to all the refinements in sea ice parametrizations absent in LIM2). Thus the question of whether a particular ITD configuration discretization or number of categories ensures a more 75 realistic sea ice variability and long-term trend remains unanswered.

Sea ice concentration (SIC) and thickness are the main quantities used to characterize itsice cover variability. Most of the previous studies have focused on the impact of an ITD on the sea ice thickness, especially in the Arctic [e.g., Holland et al., 2006; Hunke, 2014; Ungermann et al., 2017]. By contrast, SIC has received less attention, perhaps motivated by the relatively minor or only indirect effect that the ITD appears to have on the representation of itsthe mean state [e.g., Massonnet et al.,

80 2011; Uotila et al., 2017; Massonnet et al., 2019]. However, while SIC has continuously been measured by satellites since 1978 [Cavalieri et al., 1996; EUMETSAT, 2015], equivalent measurements of thickness have only become available in the past decade [e.g., Laxon et al., 2013]. Literature exploring the observed SIC variability is therefore much richer than that on sea ice thickness and offers a more exhaustive account of its key features and drivers (see most of the references above). This study therefore represents a step forward with respect to previous ones, as it presents-the, to our knowledge, the first 85 detailed assessment of the impact of the ITD discretization on the SIC variability at both poles since 1978, using the state-ofthe-art coupled ocean-sea ice model NEMO3.6-LIM3. This study is a companion paper to Massonnet et al. [2019], in which the response of the modelled sea ice climatologymean state to an ITD discretization is investigated.

The paper is structured as follows: Section 2 describes the model and experimental design,-\_Section 3 follows with the main results of the model-data comparison, and Sections 4 finishes with the discussion of the results and main conclusions.

**90 2 Model and experimental setup**

**2.1 Model description**

[revised manuscript text omitted]

**3 Results**

**3.1 Defining the winter and summer seasons**

- 160 We intend to focus the comparison between simulated and observed SIC variability in two seasons centered around winter and summer, when maximum and minimum sea ice areas occur respectively. To avoid any a priori assumption about which months define these seasons, we first assess agreement across monthly clusters and aggregate months with similar variability. Following the steps described in Section 2.3 for each observational product separately, we first calculate 3 (asthe optimal number) clusters in each individual month in the Arctic and Antarctica. At each pole, we then compute the spatial 165 correlation coefficients between all the clusters in any two months. We retain the maximum positive value from the resultant distribution, which sets the uppermost-limit of cluster agreement between those two months. Results in OSI SAF are shown in Fig. 2 (results of NSIDC and HadISST are very similar and therefore not shown). Two periods stand out at both poles, when The winter and summer seasons are then defined by finding the three months which have the largest and the immediately second largest correlation coefficients in the winter and summer half year (November-April and May-October 170 respectively). The two seasons must be and are consistent across the three observational datasets included in the analysis. This method renders two seasons in which monthly cluster agreement is largest, consistently high: January through March (JFM), and August through October (ASO). The use of JFM as the winter season is consistent with the Principal Component analysis in Close et al. (2017), in which monthly modes were best correlated in JFM as well. We find no major differences if the clusters are calculated in winter including April (JFMA) and summer including July (JASO). All the subsequent analyses
- 175 focus on thesethe two seasons\_JFM and ASO, which we refer to as winter and summer (even though they include climatological spring and fall months).

**3.2 Sea ice extent**

Before comparing SIC clusters, we explore the impact of the ITD configurationdiscretization on the temporal evolution of the Arctic and Antarctic sea ice extent (SIE) over the period 1979–2014 (Fig. 3). This analysis will help interpret results from the clusters belowpresented later. Note that impacts on the simulated climatological mean state and seasonal cycle over this period have previously been described by Massonnet et al. [2019]. In the model, seasonal SIE is calculated from monthly SIC on the original model grid; in observations, seasonal SIE is calculated from the monthly SIE directly provided by the different products. The impact of different ITD configurationsdiscretizations on the Arctic SIE in both seasons and Antarctic SIE in winter is marginal, and all the simulations show values that are within observational uncertainty (which we assume to be

- 185 defined by the envelope of the different observational products; Fig. 3). The largest differences across simulations are for the summer Antarctic SIE. Increasing the number of categories from 1 to 50 in the S1 configurations discretizations reduces the Antarctic SIE by about 4:106 km2, although the largest decrease, of about 2:106 km2, is from S1.01 tobetween S1.01 and S1.03. This renders the simulated SIE values in the S1 runscases closer to those in OSI SAF and NSIDC but more different to those in HadISST. HadISST SIE values are consistently above those in OSI SAF and NSIDC in the Arctic
- 190 and Antarctica in both seasons, as also noted by Titchner and Rayner [2014]. Increasing the number of categories in the S2 and S3 configurations discretizations has a comparatively smaller impact, reducing and increasing the summer Antarctic SIE by about 1:106 km2 respectively; these results are, yet still within observational uncertainty. The simulated SIE trend is slightly underestimated in the winter Arctic, although it is well captured in summer, as well as in Antarctica in both seasons. In terms of interannual variability, the simulations disagree the most with the observations in Antarctica, especially in summer, when the simulations show large interannual variability for all ITD in observations (for example, around 2000). By contrast, the simulated Arctic SIE variability for all ITD
- in observations (for example, around 2000). By contrast, the simu configurations discretizations is very close to the observations in both seasons.

To characterize differences between simulated and observed SIC, we calculate the integrated ice edge error (IIEE) as the total area where model and observations disagree on SIC values above 15% [Goessling et al., 2016]. In general terms, the largest IIEE is in the Arctic and Antarctica in JFA4winter, with the smallest values emerging for the comparisonwhen compared with NSIDC (Supp. Fig. 1). For all the simulations, the IIEE remains relatively constant over the period 1979–2014 at both poles and seasons, and the impact of a different ITDe IID discretization on the IIEE is marginal in the Arctic in both seasons and in Antarctica in winter. The situation is different in the Antarctic summer (JFM), when differences in IIEE due to the ITD are the largest (Fig. 4). IIEE between the simulations and observations is overall larger than across observations for all the 205 ITD configurations lincreasing the number of categories in the S1 and S3 configurationscases tends to reduce the IIEE by about 1.106 km2 between the coarsest and finest resolution. Changes in categories in the S2

configurationacross S2 discretizations have a smaller impact on the IIEE, with no clear improvement or worsening for a finer or coarser ITD. These results suggest that a finer resolution of the thinner ice and not of the thicker sea ice to some degree improves the representation of the simulated Antarctic SIC in ASOwinter in our model with respect to observations. This might be related to an improved response of the thin ice (the easiest to melt, grow, and advect) to the atmospheric forcing.

**3.3 SIC cluster analysis**

210

In the following, we describe the three clusters of SIC variability in the observations. Clusters in OSI SAF are shown in Figs. 5 and 6 in the Arctic and Antarctica respectively (since clusters in NSIDC and HadISST are very similar, they are shown in Supp. 215 Figs. 2 and 3 respectively). In the Arctic winter, the first cluster shows four poles of dominant variability, with more ice in the Barents, Greenland, and Okhotsk seas and less ice in the Labrador and Bering seas (Fig. 5); this pattern agrees with the quadrupole mode described byin previous literature associated with variations in the strength of the Siberian High [e.g., Ukita et al., 2007; Close et al., 2017]. The second cluster presents similar centers of action to the first one, but SIC anomalies are negative in the Labrador, Barents and Okhotsk seas and positive in the Bering Sea. The third cluster shows strong 220 anomalies of opposite signsigns in the Labrador (strongly positive) and Nordic seas (negative)-which, a pattern that resembles the typical fingerprint of a positive NAO phase on the SIC [Bader et al., 2011]. In fact, this cluster dominates between 1990 and 1996, when the winter NAO was persistently positive [Hurrell and Deser, 2010]. Overall, the first and third clusters alternate until 2004 approximately, after which the second cluster dominates. In the last decade, the root mean square distance between the clusters and the anomaly fields (indicated by the symbol size in Fig. 5) increases to its largest values 225 over the whole period in OSI SAF, but not in NSIDC and HadISST. These results suggest that the winter SIC variability might fundamentally have changed after 2004, in agreement with the observed acceleration in the SIC melting trend [e.g., Comiso et al., 2008; Serreze and Stroeve, 2015].

In the Arctic summer, both the cluster patterns and relative occurrences reflect a long-term melting trend (Fig. 5). The first and third clusters are very similar, which respectively; both exhibit widespread positive and negative anomalies in the central Arctic and dominate over the initial period (ca. 1979–1988) and last one (ca. 2005–2014) respectively. The second cluster, by contrast, dominates in the middle decades (ca. 1989–2005) and presents a dipole of positive and negative anomalies between the central Arctic and the surroundings. Such partitioning in decades of alternating dominance suggests that the long-term melting trend in sea ice (as seen in the SIE; Fig. 3b) controls the clustering; previously detrending the data might therefore be necessary for a more robust characterization of the interannual variability (see below).

235 In Antarcticothe Antarctic summer (JFM), the three clusters exhibit poles of dominant variability close to the continental coast, especially in the Weddell and Ross seas (Fig. 6). The first and second clusters show similar patterns but of opposite sign, with an overall decrease or increase respectively but in the Amundsen and Bellinghausen seas. The third cluster shows

dominated by the first cluster (58%), especially during the first decades. Although the second and especially the third clusters are much less frequent (31% and 11% respectively), the second one tends to dominate in the last decade (ca. 2005–2014). 240 This might be due to a slight positive trend, as seen in the SIE (Fig. 3d).

In the Antarctica\_ASO (winter), the Antarctic\_first and second clusters show opposite-sign poles in the Weddell, Bellinghausen, and Amundsen seas, with smaller contributions from other seasthe others (Fig. 6). These two modes resemble SIC variability driven by Rossby wave activ